# Determinants of institutional maternity services utilization in Myanmar

**Khaing Zar Lwin**[¤]*, **Sureeporn Punpuing***

Institute for Population and Social Research, Mahidol University, Salaya, Nakhonpathom, Thailand

¤ Current address: Mayangone Township, Yangon, Myanmar
* khaingzar.lwi@student.mahidol.ac.th, lwin.khinezar11@gmail.com (KZL); sureeporn.pun@mahidol.ac.th (SP)

## Abstract

### Background

Maternal mortality is a persistent public health problem worldwide. The maternal mortality ratio of Myanmar was 250 deaths per 100,000 live births in 2017 which was the second-highest among ASEAN member countries in that year. Myanmar's infant mortality rate was twice the average of ASEAN member countries in 2020. This study examined factors influencing institutional maternity service utilization and identified the need for improved maternal health outcomes.

### Methods

A cross-sectional study design was used to examine the experience of 3,642 women from the 2015–16 Myanmar Demographic and Health Survey by adapting Andersen's Behavioral Model. Both descriptive and inferential statistics were applied. Adjusted odds ratios and 95% confidence interval were reported in the logistic regression results.

### Results

The findings illustrate that the proportion of women who delivered their last child in a health/clinical care facility was 39.7%. Women live in rural areas, states/regions with a high levels of poverty, poor households, experience with financial burden and the husband's occupation in agriculture or unskilled labor were negatively associated with institutional delivery. While a greater number of ANC visits and level of the couple's education had a positive association with institutional delivery.

### Conclusion

The determinants of institutional delivery utilization in this study related to the institutional facilities environment imply an improvement of the institutional availability and accessibility in rural areas, and different states/regions, particularly Chin, Kayah and Kachin States- the poorest states in Myanmar. The poverty reduction strategies are urgently implemented because problems on health care costs and household economic status played important roles in institutional delivery utilization. The ANC visits indicated a significant increase in

**Data Availability Statement:** All relevant data are within the manuscript and its Supporting information files. Data using in this study are available without restriction. However, a person is necessary to register in DHS program to access

complete data set. We can distribute our syntax file for data analysis. 1) DHS data can be assessed on: https://dhsprogram.com/data/available-datasets.cfm and can be registered on https://dhsprogram.com/data/new-user-registration.cfm. 2) The data set and questionnaires used for our study can be accessed on: https://dhsprogram.com/data/dataset/Myanmar_Standard-DHS_2016.cfm?flag=0.

**Funding:** The author(s) received no specific funding for this work.

**Competing interests:** The authors have declared that no competing interests exist.

**Abbreviations:** ANC, Antenatal Care; aOR, Adjusted Odds Ratio; CI, Confidence Interval; MCH, Maternal and Child Health; MDHS, Myanmar Demographic and Health Survey; MMR, Maternal Mortality Rate; RHC, Rural Health Center; SRHC, Sub-Rural Health Center; UNICEF, United Nations Children's Fund; WHO, World Health Organization.

institutional delivery. The government needs to motivate vulnerable population groups to seek ANC and institutional delivery. Moreover, education is crucial in increasing health knowledge, skills, and capabilities. Thus, improving access to quality, formal, and informal education is necessary.

## Introduction

Pregnant women often have a risk for complications during pregnancy, delivery, and the early post-partum period. When women deliver at a healthcare facility, obstetrical complications can be managed in a timely way, and there is usually sufficient equipment and skilled health personnel on hand. It is well-recognized that child delivery in a health/clinical institution under the care of trained healthcare personnel promotes the survival of the mother and new-born, and reduces the risk of complications and maternal mortality. Furthermore, institutional delivery is one of the key interventions to reduce maternal mortality and morbidity [1, 2].

Worldwide, the number of women who die from pregnancy-related complications and childbirth has dropped from 451,000 in 2000 to 295,000 in 2017 [3]. Most of the maternal deaths (94%) occur in developing countries where there is low accessibility and utilization of maternity services [4]. In 2019, there were an estimated 1.9 million stillbirths globally and, of these, 84% of stillbirth occurred in low and lower-middle-income countries [5]. About 70% of deliveries by lower-income women in sub-Saharan Africa, South Asia, and Southeast Asia occur at home [6].

Myanmar is one of the Southeast Asian countries and had an estimated population of 51.1 million in 2019. The majority of the Myanmar population (70%) lives in rural areas [7]. The maternal mortality ratio (MMR) of Myanmar declined from 340 in 2000 to 250 deaths per 100,000 live births in 2017 [3]. Although the MMR has declined, Myanmar has a long way to go in order to meet the 2030 Sustainable Development Goal for the MMR (70 /100,000 live births). Moreover, the MMR of Myanmar was the highest among Southeast Asian countries in 2017 and nearly twice the regional average of 137 [3]. In 2014, 38.5% of maternal deaths in Myanmar occurred six weeks after delivery, and 32.4% occurred during delivery [8]. Moreover, the infant mortality rate (IMR) of Myanmar was 38 per 1,000 live births in 2020, or twice the regional average [9]. There are proven healthcare interventions to prevent or manage pregnancy complications, including antenatal care (ANC), delivering at a health/clinical facility, and postpartum care six weeks after delivery. However, millions of mothers in Myanmar remain at risk during pregnancy and delivery due to the inability to afford healthcare costs, and/or difficulty accessing a facility and skilled birth attendant [10].

In 2007, only one in four (23.7%) deliveries took place in a healthcare facility in Myanmar [11]. In 2010, 36.2% of women were delivered in a healthcare facility [12]. Six years later, in 2016, the rate of institutional delivery increased to only 37.0%, and this was the second-lowest among Southeast Asian countries [13]. In the ASEAN member countries of Singapore, Brunei, and Thailand, institutional delivery was 100% in 2004, 2009, and 2012, respectively. With the exception of Myanmar and Laos, institutional delivery rates in other ASEAN member countries all exceed 70% [14].

Pregnant women in rural areas commonly seek maternal healthcare from the local health centers. In 2018, there were 13,594 village tracts located in seven states (Kachin, Kayah, Kayin, Chin, Mon, Rakhine, and Shan), seven regions (Tanintharyi, Sagaing, Magway, Bago, Yangon, Mandalay, Ayeyarwady), and one Union Territory (Nay Pyi Taw). The number of non-

municipal health centers was 1,796 [15]. The maternal mortality level of the rural population in Myanmar was higher than their urban counterparts (310 vs 193), according to the 2014 Myanmar population and housing census [8]. In 2017, 76% of all maternal deaths occurred in rural areas and 23% occurred in urban areas [16]. As high as 90% of women in urban areas received at least one ANC exam by a skilled health care provider, compared with a maximum of 78% for women in rural areas [11–13]. The utilization of institutional delivery in urban areas was 51% in 2007 [11], 65% in 2010 [12] and 70% in 2016 [13]. In rural areas, the utilization of institutional delivery was 15% in 2007, which is more than three times lower than that of urban areas [11], although the level had increased to 25% by 2010 [12] and to 28% by 2016 [13].

The Myanmar Ministry of Health is implementing a new strategy for improving maternal and child health (MCH) by shifting the focus from pregnancy to delivery, and from home to institutional delivery [17]. Healthcare facilities in rural Myanmar are sparsely distributed, and this reduces physical access to maternal care [18]. In addition, rural Myanmar women are mostly lower-income and cannot easily afford transportation to a clinical facility for childbirth [19]. In 2017, nearly one in four Myanmar citizens were living below the national poverty line [20]. Moreover, rural health centers have insufficient medical equipment and medical supplies for obstetric care and are frequently understaffed [9]. Moreover, it was found that there are differences in health accessibility among population in different states/regions in Myanmar. In 2018, there were 1134 public general hospitals in Myanmar, in which the availability of healthcare facilities especially public hospital resources is highest in the Ayeyarwady Region and lowest in Kayin State, followed by Chin State. As of the private sector, 253 private hospitals were distributed healthcare services, in which Yangon Region is highest availability of private healthcare services, followed by Mandalay Region. No private hospital is located in Kayah State [21].

Most previous studies about institutional delivery in Myanmar were conducted in only a few locations of Myanmar. By contrast, this study used data from a national sample of reproductive-age women to analyze the institutional facility availability and accessibility, the need for maternity care, and the enabling and predisposing factors which affect the decision to deliver in a health/clinical facility.

## Andersen's model of health service utilization

The analytical design for this study is based on Andersen's behavioral model of health services utilization. Myanmar is a developing country, and the government is trying to strengthen the overall health system, especially by improving MCH status. Andersen's model suggests certain proxy determinants of institutional maternity service utilization, and understanding how these factors interact can help inform programs that are trying to increase institutional delivery. Andersen developed the model in the late 1960s in order to improve understanding of health services utilization among families, to describe and measure the equitable access of healthcare services, and to support policy formation for equitable access of health services. In the 1990s, Andersen modified earlier versions of the model by adding or refining factors related to the service environment, certain population characteristics, health behaviors, and outcomes [22].

The healthcare system (resources for healthcare, availability, affordability, accessibility to healthcare services) and external environment (the political and economic situation of the country) are under the component of 'environment' in the model. Under population characteristics, need-based characteristics include the severity of illness/condition and requirement for healthcare intervention. Enabling characteristics to include family- and community-level variables and the predisposing factors are socio-demographic characteristics at the individual level. The health behavior factor encompasses personal health practices such as diet and exercise [22].

Studies from developed and developing countries have applied the Andersen model, such as the United States [23], Nepal [24, 25], Nigeria [26], and Pakistan [27] as a framework for conducting research about health service utilization. Thus, it can be expected that the Andersen model is also applicable to a country like 'Myanmar' which is not radically different from other developing countries in the region (e.g., Nepal). This study adapted the 'environment' factor in Andersen's model to focus on the "Institutional Delivery" in order to explore the influence of the availability, accessibility, and affordability of giving birth at a healthcare facility in Myanmar. Regarding need-based characteristics, women who perceive the need for professional help and are aware of the risks of pregnancy and delivery are expected to seek antenatal care (ANC) and prepare for delivery in advance. Risk assessment could be conducted and women may be advised to deliver at the health facility if abnormal conditions of pregnancy are found during the ANC visit, such as twin pregnancy. Having four or more ANC exams may reflect a woman's concern about her pregnancy, experience of danger signs of pregnancy, and the need for professional help [28–30]. Therefore, ANC visits and experience of pregnancy complications can influence a woman's choice of place of delivery. In enabling characteristics, occupation of woman and husband, and household wealth status are potential determinants of the decision to give birth in a health/clinical facility. Age of woman at last delivery and the couple's education were selected as predisposing factors in this study because they are hypothesized to have a significant influence on the utilization of a healthcare facility. The conceptual framework that adapted Andersen's Model of this study is presented in Fig 1.

## Methods

### Data and sampling design

The data for this study was extracted from the 2015–16 Myanmar Demographic and Health Survey (MDHS) [31]. The MDHS is the first nationwide survey of the population and health

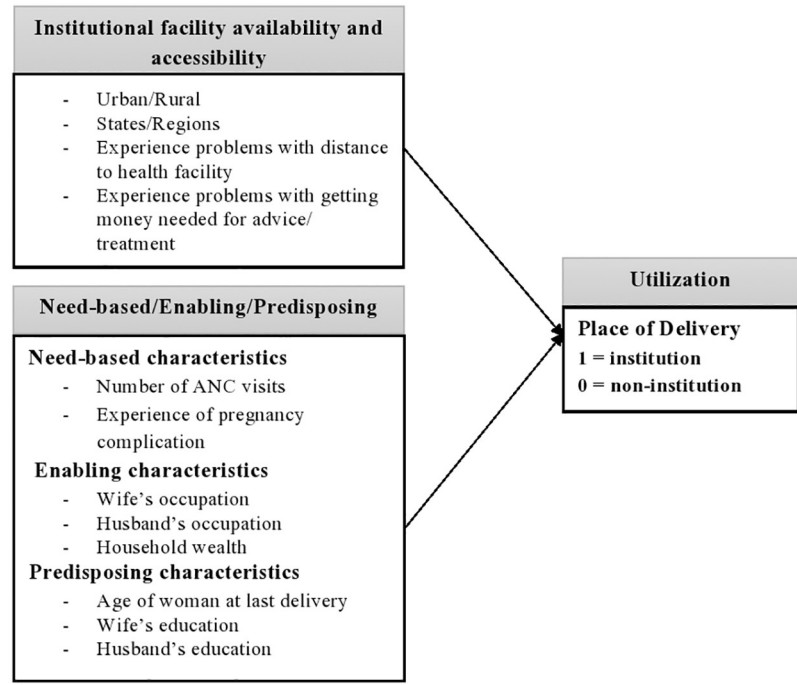

**Fig 1. Conceptual framework that adapted Andersen's model for institutional maternity service.**

status in Myanmar. A two-stage stratified sampling method was applied in the MDHS. The first stage sampled clusters from village tracts using probability proportional to the size that included 123 clusters from urban areas, and 319 clusters from rural areas. The second stage employed systematic probability sampling with a quota of 30 households from each cluster. The final, nationally representative sample was 12,500 households. A total of 13,454 women were eligible to participate in the MDHS, and 12,885 women (age 15–49 years) successfully completed the interview. Among these women, mothers who delivered their last child within five years preceding the survey were selected for this study. Visitors/non-residents of the household and women who did not answer the place of their last childbirth were excluded from the sample. In order to avoid bias of household information in the analysis, only one eligible woman per household was randomly selected. After considering these four criteria, 3,642 ever-married women were included. Individual sampling weights were applied in all analyses for this study. After weighting, the final sample totaled 3,383 ever-married women for this study (S1 Fig).

## Outcome variable

**Institutional delivery.** The outcome variable in this study is a place of delivery for the last child born within 5 years prior to the survey. The response for this variable was dichotomized as "institutional" and "non-institutional". 'Institutional delivery' means a delivery that took place in a government or private health/clinical facility under the overall management of trained healthcare providers, and includes public healthcare providers (e.g., government hospitals, rural health centers [RHCs] which are located in rural areas, and no inpatient. At the RHC, there are four staff, which are a health assistant, a lady health visitor, a midwife, and a public health supervisor grade I., urban health centers [UHCs], mobile clinics, and MCH centers), private hospitals/clinics, private maternity homes, other private medical providers, and NGO healthcare providers. Women who delivered their last child outside an equipped healthcare facility were considered to have had a 'non-institutional delivery' [13].

## Exposure variables

The exposure variables were divided into 'institutional facility availability and accessibility' and 'population characteristics' based on Andersen's healthcare utilization model [22]. Regarding the institutional facility availability and accessibility, many studies have found that the chance of institutional delivery by trained health personnel is lower for women who live in a rural area because of less convenient or affordable access to a healthcare facility, and norms that may encourage home delivery by a traditional birth attendant (TBA) [19, 25, 32, 33]. Moreover, other studies have identified the cost of travel and care as barriers to institutional delivery [19, 32, 34]. Another study in Myanmar found that the frequency of ANC visits is a predictor of institutional delivery [18, 19]. The components of population characteristics and enabling and predisposing factors are also potential determinants of institutional delivery [26, 35–37]. The detailed description of variables for this study is shown in S1 Table.

## Statistical analysis

Data were analyzed by applying STATA version 14 statistical software [38]. Descriptive, bivariate, and binary logistic regression analysis was conducted to explore the statistical association of factors with institutional maternity service utilization. Individual sample weights were applied for all analyses to compensate for the complexity of study design and unequal selection probabilities. The Chi-square test was applied to examine the association between exposure variables and institutional delivery. The test for multicollinearity was conducted by using

Spearman's correlation in order to confirm the existence of multicollinearity among the variables (S2 Table) [39]. Two models were developed in the binary logistic regression by applying Andersen's model where the odds ratios were within the 95% confidence interval (CI). The level of statistical significance was set at a p-value <0.05.

### Ethics approval

Ethical approval was not required because of the availability of a dataset from the DHS website (https://dhsprogram.com/) after registering on the DHS website. This study was deemed exempt by the Institutional Review Board, Institute for Population and Social Research (IPSR-IRB).

## Results

### Description of sample and status of institutional delivery

In this sample, nearly two out of five women (39.7%) delivered their last child at a healthcare facility (i.e., institution). More than three in four (77.3%) women lived in a rural area. The highest proportion of the sample women was in the Ayeyarwady Region (13.8%), and the lowest proportion was in the Kayah State (0.7%). There were 27.5% and 38.9% indicated experiences problems with distance to health facility, and getting money needed for advice/treatment respectively. More than half (58.4%) of women had at least 4 ANC visits. About one-tenth (11.7%) have had a previous pregnancy complication. About one-third of women (34.9%) were not working. Husbands' occupation was commonly in unskilled manual labor (38.6%). About half (49.4%) of the sample lived in low economic-status households (poor = 22.7%, and poorest = 26.7%). More than half of women were in 25–34 age group, 46.1% of the sample attained primary education, and 16.6% had no education. Two out of five women (41.1%) reported that their spouse completed primary school, and 16.6% had no education (S3 Table). The bivariate analysis results for this study is presented in S4 Table.

### Binary logistic regression of institutional delivery with environment and population characteristics

Table 1 presents the net effects of institutional facility availability and accessibility and population characteristics on the place of delivery for this sample of Myanmar women. Two models were constructed based on Andersen's conceptual framework. In Myanmar, rural residents seek ANC and maternity services from RHCs and SRHCs. Urban residents have more options given the greater prevalence of public and private health centers/clinics and MCH centers. In an emergency, rural residents are referred to 'station' hospitals which are the smallest hospital (16 or 25-bed hospital) located in rural areas, and there are two doctors, six nurses, two technicians, and seven auxiliary staff and urban residents are referred to the township hospital (at least 50-bed hospital) [17]. Starting in 2014, the Myanmar government gave greater priority to increasing the health budget and providing essential health care at no cost. However, there have been reports of some public healthcare providers requiring out-of-pocket payment for ANC and delivery [40]. Therefore, variables in the first model are mainly focused on the institutional facility availability and accessibility. The second model included all characteristics in Andersen's behavioral model of health service utilization.

All factors under the institutional facility availability and accessibility variable were significantly associated with the utilization of institutional delivery in Model 1. Before controlling for population characteristics, women who lived in a rural area were 79% less likely to deliver at an institution compared to those who lived in urban. Women who lived in different states/

**Table 1. Adjusted Odds Ratios (aOR): Factors associated with institutional delivery *(N = 3268)*.**

| Exposure variables | Model 1 | | Model 2 | |
|---|---|---|---|---|
| | aOR | 95% CI | aOR | 95% CI |
| **Institutional facility availability and accessibility** | | | | |
| **Urban/Rural** | | | | |
| Urban | Ref | | Ref | |
| Rural | 0.21*** | 0.14–0.19 | 0.32*** | 0.22–0.46 |
| **States/Regions** | | | | |
| Yangon | Ref | | Ref | |
| Kayah | 0.40** | 0.20–0.80 | 0.37*** | 0.20–0.68 |
| Kayin | 0.83 | 0.41–1.70 | 1.24 | 0.64–2.43 |
| Chin | 0.24*** | 0.12–0.45 | 0.25*** | 0.13–0.49 |
| Sagaing | 0.46* | 0.23–0.90 | 0.57 | 0.29–1.11 |
| Tanintharyi | 0.82 | 0.43–1.57 | 0.98 | 0.51–1.86 |
| Bago | 0.58 | 0.31–1.06 | 0.71 | 0.40–1.26 |
| Magway | 0.66 | 0.35–1.27 | 0.79 | 0.41–1.50 |
| Mandalay | 0.68 | 0.37–1.26 | 0.74 | 0.40–1.38 |
| Mon | 0.47* | 0.24–0.90 | 0.55 | 0.30–1.01 |
| Rakhine | 0.31*** | 0.16–0.62 | 0.61 | 0.33–1.13 |
| Kachin | 0.47* | 0.24–0.93 | 0.48* | 0.26–0.88 |
| Shan | 0.43* | 0.21–0.90 | 0.70 | 0.34–1.42 |
| Ayeyarwady | 0.59 | 0.31–1.13 | 0.79 | 0.41–1.51 |
| Nay Pyi Taw | 0.59 | 0.32–1.12 | 0.89 | 0.48–1.67 |
| **Experience problems with distance to health facility** | | | | |
| No | Ref | | Ref | |
| Yes | 0.68** | 0.51–0.90 | 0.91 | 0.68–1.22 |
| **Experience problems with getting money needed for advice/ treatment** | | | | |
| No | Ref | | Ref | |
| Yes | 0.51*** | 0.41–0.63 | 0.76* | 0.60–0.97 |
| **Need-based Characteristics** | | | | |
| **Number of ANC visits** | | | | |
| No ANC visit | | | Ref | |
| 1–3 times | | | 3.93** | 2.33–6.62 |
| 4 times and more | | | 7.20*** | 4.31–12.03 |
| **Experience of pregnancy complication** | | | | |
| No | | | Ref | |
| Yes | | | 1.08 | 0.80–1.48 |
| **Enabling Characteristics** | | | | |
| **Wife's occupation*** | | | | |
| Managerial/professional | | | Ref | |
| Agriculture | | | 1.11 | 0.62–2.01 |
| Skilled manual | | | 1.04 | 0.62–1.76 |
| Unskilled manual | | | 1.17 | 0.67–2.03 |
| Not working | | | 1.19 | 0.70–2.03 |
| **Husband's occupation** | | | | |
| Managerial/professional | | | Ref | |
| Agriculture | | | 0.52* | 0.31–0.86 |
| Skilled manual labor | | | 0.71 | 0.44–1.14 |
| Unskilled manual labor | | | 0.61* | 0.38–0.97 |

*(Continued)*

**Table 1.** (Continued)

| Exposure variables | Model 1 | | Model 2 | |
|---|---|---|---|---|
| | aOR | 95% CI | aOR | 95% CI |
| **Household wealth** | | | | |
| Wealthier | | | Ref | |
| Average | | | 0.72* | 0.54–0.95 |
| Poorer | | | 0.46*** | 0.34–0.62 |
| **Predisposing Characteristics** | | | | |
| **Age of woman at last delivery** | | | | |
| < = 24 | | | Ref | |
| 25–34 | | | 0.92 | 0.72–1.17 |
| 35+ | | | 1.26 | 0.96–1.66 |
| **Wife's education** | | | | |
| No education | | | Ref | |
| Primary | | | 1.45* | 1.02–2.06 |
| Secondary | | | 2.25*** | 1.51–3.35 |
| Tertiary | | | 3.31*** | 1.83–5.99 |
| **Husband's education** | | | | |
| No education | | | Ref | |
| Primary | | | 1.37 | 1.00–1.89 |
| Secondary | | | 1.37 | 0.98–1.90 |
| Tertiary | | | 2.76*** | 1.54–4.95 |
| Constant | 5.43*** | 3.34–8.83 | 0.44 | 0.18–1.10 |
| F-test | 13.27*** | | 10.8*** | |

Note: Significant level

\* = p<0.05;

\*\* = p<0.01;

\*\*\* = p<0.001.

regions were 53% to 76% less likely to deliver at the health facilities, compared to those living in Yangon Region. Moreover, women who experienced problems with distance to health facility and getting money needed for advice/ treatment were 32% and 49% less likely to use institutional delivery than those who did not experience such problems, respectively.

The odds ratios in Model 2 declined from the level in Model 1 after controlling for population characteristics; only urban/rural, states/regions and experience problems with getting money needed for advice/ treatment under the institutional facility availability and accessibility were significant determinants of the utilization of institutional delivery. The number of ANC visits, occupation of husband, household wealth status, and education of the woman and husband were significantly associated with utilization of institutional delivery after controlling for each other.

Concentrating on the availability and accessibility of healthcare services, women who lived in rural were 68% less likely to use institutional delivery than those living in urban. Women who lived in Chin, Kayah, and Kachin States were 75%, 63%, 52% less likely to give birth at health facility respectively, compared to those living in Yangon Region. Women who experienced problems with getting money needed for advice/ treatment were 24% less likely to use institutional delivery than women who did not experience problems with getting money needed.

For the need-based factors, women who had at least 4 or 1–3 ANC visits were 7.2 and 3.9 times more likely to use institutional delivery, respectively, compared to those who had no ANC visit. For enabling characteristics, women whose husbands worked in the agriculture sector and unskilled manual labor were 45% and 39%, respectively, less likely to use institutional delivery than those whose husbands worked in managerial/professional jobs. Women who lived in a household with average economic status or a less well-off household were 28% and 54%, respectively, less likely to deliver at healthcare facilities than those who lived in a wealthier household. For predisposing characteristics, women with tertiary, secondary and primary level education were 3.3, 2.3, and 1.5 times, respectively, more likely to use institutional delivery compared to women with no education. Women whose husbands had tertiary education were 2.8 times more likely to deliver at a healthcare facility than those whose husbands had less or no education.

## Discussion

This study examined the factors associated with institutional delivery in Myanmar by applying Andersen's model of health service utilization. In this sample of reproductive-age women, more than half of the most recent infant deliveries occurred outside of a healthcare facility. The availability and accessibility of health services, the number of ANC visits, husband's occupation, household wealth status, and education of the woman and her husband were significant predictors of institutional maternity service utilization.

Place of residence (urban/rural and states/regions) related to accessibility and availability of quality healthcare services, and socio-economic status of the household. In this study, women in rural areas had significantly lower institutional delivery than their urban counterparts. Many studies have established that the availability of institutional delivery is lower in rural areas [19, 26, 32, 41–43]. Similarly, a study conducted in the Magway Region, a dry zone [44] of Myanmar, found that women who lived near a health center with easy access to a delivery room were more likely to use institutional delivery [18]. The MMR in rural areas was 1.6 times higher than in urban areas [9]. Moreover, 70% of the total population in Myanmar are rural residents [7] and most rural residents seek healthcare from the station hospitals, RHCs, and SRHCs which are the smallest healthcare unit in rural areas. With its 13,594 village tracts and 63,276 villages, Myanmar had a total of 746 station hospitals, 1,796 RHCs [15] and 9,152 SRHCs as of 2018. The ratios of health facilities to 1,000 rural population are as follows: 0.02 for station hospitals, 0.05 for the RHC, and 0.25 for the SRHC [45]. Therefore, inadequate service infrastructure can make it difficult for rural women to deliver at a healthcare facility. Furthermore, in 2016, the health force density of Myanmar was 2.4 healthcare providers per 1,000 population, and that reflects the unequal distribution of health professionals based on the total population of the country [46]. According to World Health Organization (WHO), an adequate health force density should be at least 4.4 healthcare providers per 1,000 population [47]. Myanmar is short of human resources for health, mainly due to a difference between supply and demand for health professionals.

This study indicated that women in Chin, Kayah, and Kachin states were 75%, 63%, 52% less likely to deliver at the health facility respectively than those living in Yangon Region. And those who experienced problems with getting money needed for advice/ treatment were 24% less likely to use institutional delivery than women who did not experience such problems. The Myanmar living condition survey 2017 also found that the regions/states had an influence, on the different levels of health care accessibility, particularly related to the population's financial burden [20]. In addition, based on the poverty headcount index, Chin state had the highest (58.0%) population living below the poverty line in Myanmar, while Kayah and Kachin States

also had 32.2% and 36.6% respectively population live below the poverty line [20]. This study confirms that financial problems play an important role in determining the use of institutional health delivery.

The women's experienced problems with distance to a healthcare facility and getting money needed for advice /treatment at health facility were statistically negative associated with utilization of institutional delivery in Model 1, but the experience problems with distance to health facility was insignificant in Model 2 due to the interaction with other variables. Most previous studies found that utilization of healthcare facilities for maternity service is significantly influenced by distance to healthcare facility in that, as the distance increased, utilization of maternity services decreased [34, 48]. One study conducted in a rural area of the Magway Region found that long-distance to a healthcare facility is one of the major deterrents for women to seek ANC and delivery services [18]. Another study in central Myanmar also found that the cost of service, distance, and lack of transportation were the main reasons why women did not deliver at a healthcare facility [19]. Even though women in Myanmar can get free maternal healthcare services from any public service provider, there are reports of some government facilities requiring out-of-pocket payments. Other things being equal, the cost of institutional delivery is seven times more than that of home delivery [41].

The number of ANC visits was also a strong positive predictor of institutional delivery. Pregnant women who had at least four ANC visits had significantly higher odds of utilizing institutional delivery. Similar results have been found in previous studies [19, 35, 37, 42, 43, 49]. During the ANC visit, high-risk women are referred to a larger healthcare facility to receive more sophisticated care to reduce the risk of complications of childbirth. One study in the rural area of a township in the Magway Region found that women who received ANC from a trained provider were more likely to deliver at a healthcare facility [18]. Another study in Myanmar found that women who did not have ANC were 6.1 times more likely to deliver at home compared to those with 4 or more ANC visits [50]. One study in Myanmar found that there was inequality of ANC service utilization among adolescent pregnant women [51], and some pregnant women preferred using the local TBA for ANC and delivery [52]. WHO recommends that pregnant women should receive at least 4 ANC check-ups by a trained provider, and 2016 WHO ANC model recommends that 8 ANC check-ups are ideal to promote optimal pregnancy experiences [28]. Women who received counseling about the potential for pregnancy complications may be more motivated to attend ANC check-ups with a trained health provider and institutional delivery [49]. Women who experienced pregnancy complications may have high utilization of institutional delivery. A study conducted in Nepal found that women who reported pregnancy complications such as bleeding had a high odds of institutional delivery [24]. Women may experience more complications of pregnancy as they get older. Similarly, multiparous women may also experience a higher prevalence of pregnancy complications. Consequences of previous births or miscarriages may make women more aware of the risks of childbirth and the benefits of delivery at a health facility [29].

Specifically, this study found that women whose husbands were employed in agriculture and unskilled manual jobs were less likely to deliver at a healthcare facility compared to those whose husbands were worked in a managerial/professional position. That finding is consistent with previous studies [37, 53]. As in lower-middle-income countries where families are struggling just to survive, healthcare might be seen as a luxury compared to other basic needs such as food and shelter. Also, in traditional Myanmar culture, the male head of the household is considered the primary breadwinner for the family, while women are expected to perform household chores like cleaning, cooking, and caring for dependents. That said, some women help with the family business, which may be farming or cottage industry, but these are usually unpaid family jobs. In extreme cases, some women work outside the home for extra income to

address financial needs [54]. The point is that it is unrealistic for the lower-income pregnant woman to seek ANC and institutional delivery. This is because the extra expenditure would threaten her family's economic situation.

The odds of the utilization of institutional delivery were lower among women who lived in average or poorer household wealth than those with wealthier households. The more economically disadvantaged women usually live in rural areas. These findings are consistent with many previous studies [35, 37, 41–45]. It is suggested that lower-income households prioritize spending on food and housing over healthcare.

Educated women were more likely to give childbirth at a healthcare facility. This finding is consistent with several previous studies because education may enhance a woman's ability to make independent decisions [19, 32, 34, 37, 42, 43, 50]. In Myanmar, 7.3% of people over age 25 years had attained at least a university education. What is more, nearly two-thirds of the university graduates were women [55]. Education improves the ability of a woman to afford institutional healthcare, and she is likely to be more conscientious about prevention (i.e., ANC) and being delivered by someone trained in obstetrics [44]. A higher level of a woman's education should lead to more exposure to information about pregnancy, care during childbirth, risk factors, and potential delivery complications. In addition, the results of the 2010 Multiple Indicator Cluster Survey (MICS) in Myanmar found that the utilization of facility-based delivery was higher among educated women (54%) compared to only half that (25%) for women with primary education [12].

Husband's education was also an influence on woman's utilization of institutional delivery. The findings from this study are consistent with other studies [37, 43]. One study in Myanmar found that a woman whose husband is better educated had more maternal healthcare knowledge, and the more-educated husbands tended to take more care of their spouse during pregnancy and the time before and after delivery [54]. Education also increases people's awareness about life around them and encourages meaningful participation in healthcare and development. Moreover, the social and familial context in Myanmar is still mostly male-dominated [54]. Thus, a husband's education is essential in considering and encouraging his spouse to deliver at a healthcare facility.

In the past, MOHS concentrated the MCH and Reproductive Health issues under the five-year Strategic Plan for Reproductive Health (2004–2008, 2009–2013, and 2014–2018) by providing easily accessible to modern contraception, increasing skill health service providers, and promoting ANC services for pregnant women. In 2017, maternal death surveillance and response systems were implemented in every state and region with the purpose of reducing maternal mortality by improving the quality of care and investigating the causes of maternal mortality [16]. The 2017–2021 Strategies towards Ending Preventable Maternal Mortality (EPMM) in Myanmar was approved in 2018. In 2020, MOHS produced "the clinical management guideline for COVID-19 infection in pregnancy", in which all health care units that provided maternal health care services deliver a hotline or contact number to address patient concerns or inquiries [56]. At present, the 2021–2025 National Strategic Plan for sexual, reproductive, maternal, newborn, child and adolescent health (SRMNCAH) and action plans were scheduled to begin in the first quarter of 2021, delayed due to unstable political crisis [57].

However, in 2021, most civil servants, including government health professionals, participated in the Civil Disobedience Movement in which healthcare professionals protested by not attending work since the February 2021 coup. The immediate impact was insufficient staffing, and impaired ability to provide essential health care. Therefore, the accessibility of healthcare services in Myanmar is deficient [58]. Although the Ministry of Health and Sports (MOHS) had urged healthcare professionals to resume their responsibilities, it is unclear how many of them have returned to work [57]. Moreover, the armed conflict has continued in Kayah, Chin,

Shan, and the Kachin States, and Sagaing Region that has created thousands of internally displaced persons (IDPs), including pregnant women and newborns [59]. World Health Organization has supported MOHS in the implementation of early warning, alerts, and response system (EWARS) for IDPs via mobile clinics in conflict-affected areas, especially in Kachin and Rakhine States. The service implementation areas declined from 57 locations with 7,998 consultations to 24 locations with 1,282 consultations in the three months after the coup [57]. In Chin State, one pregnant woman, two newborns, and three elderly people died due to a lack of healthcare services while fleeing during the fighting [60].

After the coup, the curfew (10:00 PM to 4:00 AM) restricted the movement of people, and that posed a major obstacle for emergency obstetric referral service [57]. The curfew also certainly adversely affected accessibility to ANC, delivery, and reproductive health services due to the interruption of the transportation schedules for required supplies at health facilities. What is more, Myanmar's gross domestic product (GDP) fell by 18 percent in the 2021 Fiscal Year, and the economy has deteriorated due to the dual shocks of the coup and the Covid-19 pandemic. The main breadwinners of many households have lost their jobs, or are suffering from reduced income and high commodity prices [61]. These adverse effects are sure to worsen in the months ahead, and that will continue to negatively impact the utilization of health care services and, in particular, institutional delivery.

## Strengths and limitations

A major strength of this study is that the data come from a nationally representative sample of reproductive-age women in Myanmar. In addition, the MDHS provides data for several of the key variables in Andersen's model. There is a secure data quality as the MDHS is part of an internationally verified tool for population measurement. However, the MDHS is a cross-sectional study and, thus, causal inferences between independent and dependent variables cannot be made. Other limitations are recall bias, unmeasured confounders such as the amount of cost at healthcare facilities, and distance to healthcare facilities that were not measured in the 2015–16 MDHS. Ethnicity and religion are closely linked to cultural norms and are thought to impact beliefs and values related to childbirth and health service utilization [29]. Nevertheless, these variables are not available in the MDHS dataset.

## Conclusions

This study identified many factors that are determinants of institutional delivery among ever-married women aged 15–49 years old, and also revealed an inequality in health care utilization in Myanmar. The availability and accessibility of health services were a strong predictor of utilization of institutional delivery. The healthcare facilities can be improved by expanding the health infrastructure in rural areas, and different states/regions, particularly Chin, Kayah, and Kachin States. The unaffordable health care costs and low household wealth are significantly associated with low institutional delivery utilization. Therefore, the poverty reduction strategies through existing programs, such as increased financing for livelihood development, providing agriculture loans, livestock breeding loans, livelihood support, and encouraging voluntary contributions through corporate social responsibility should be strengthened and urgently implemented.

Moreover, the significant frequency of ANC contacts was affecting a woman's decision to give birth at a healthcare facility suggests that safe motherhood programs should emphasize the education and communication content of the ANC services. Education plays a major role in providing individuals with knowledge, skills, and capabilities to participate effectively in society. Thus, improving access to quality, formal, and informal education is necessary.

## Supporting information

**S1 Fig. Sampling cases from the 2015–16 MDHS.**
(TIF)

**S1 Table. Operational definition and measurement of variables.**
(PDF)

**S2 Table. Correlation matrix.**
(PDF)

**S3 Table. Description of sample characteristics and utilization of institutional delivery** *(N = 3383).*
(PDF)

**S4 Table. Bivariate analysis result** *(N = 3383).*
(PDF)

**S1 Data.**
(DO)

## Acknowledgments

We are gratified to the DHS program for permitting us to use 2015–2016 MDHS data for this analysis. We would like to express our special gratitude and thanks to Assistant Professor Dr. Malee Sunpuwan, Institute for Population and Social Research, Mahidol University for her advice in the conceptualization of this study.'

## Author Contributions

**Conceptualization:** Khaing Zar Lwin, Sureeporn Punpuing.

**Formal analysis:** Khaing Zar Lwin.

**Methodology:** Khaing Zar Lwin, Sureeporn Punpuing.

**Software:** Khaing Zar Lwin.

**Writing – original draft:** Khaing Zar Lwin.

**Writing – review & editing:** Sureeporn Punpuing.

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
