## [Decision Letter · Decision Letter 0]

22 Dec 2021

PONE-D-21-29455Determinants of institutional maternity services utilization in MyanmarPLOS ONE

Dear Dr. Lwin,

Thank you for submitting your manuscript to PLOS ONE. After careful consideration, we feel that it has merit but does not fully meet PLOS ONE’s publication criteria as it currently stands. Therefore, we invite you to submit a revised version of the manuscript that addresses the points raised during the review process.

We look forward to receiving your revised manuscript.

Kind regards,

Kannan Navaneetham, PhD

Academic Editor

PLOS ONE

Journal Requirements:

Reviewers' comments:

Reviewer's Responses to Questions

**Comments to the Author**

1. Is the manuscript technically sound, and do the data support the conclusions?

Reviewer #1: Yes

Reviewer #2: Yes

2. Has the statistical analysis been performed appropriately and rigorously? 

Reviewer #1: Yes

Reviewer #2: Yes

3. Have the authors made all data underlying the findings in their manuscript fully available?

Reviewer #1: Yes

Reviewer #2: Yes

4. Is the manuscript presented in an intelligible fashion and written in standard English?

Reviewer #1: Yes

Reviewer #2: Yes

5. Review Comments to the Author

Reviewer #1: The manuscript written on "Determinants of institutional maternity services utilization in Myanmar" is very useful for researchers, policy planners and students as it provides a key indicator of child and maternal health in Myanmar.

Reviewer #2: The authors have used secondary DHS data to understand the factors influencing institutional delivery in Myanmar. Overall, this is a well-written paper on a very timely and relevant topic. Understanding enabling factors and barriers to accessing maternal health services can be crucial to improvements in maternal and child health outcomes.

Following are my key observations:

Abstract

- Please note the direction of the effect of household wealth, education and husband’s occupation on institutional delivery.

Background

- The introduction has been well-written with a focus on global maternal mortality, followed by narrowing down to Myanmar. However, in line 71, there is a switch to discussing barriers to accessing maternal health services among the rural population. It might be useful to provide some context here, such as, information on maternal mortality rate among rural women, disparities in utilization of maternal health services between rural and urban areas. If one of the main aims of this study is to understand disparities in access to maternal health services between rural and urban areas, it might be helpful for the authors to use this paragraph to highlight that aspect.

Methods

- Line 135, it is unclear why only married women were included in the sample.

- Line 172, there is insufficient justification for why age and women’s occupation were chosen to be removed from the analysis. For instance, why would the authors not remove parity and husband’s occupation? Age is an important pre-disposing factor and determinant of institutional delivery.

- As a reviewer, I am slightly skeptical of the inclusion of perception-based variables within the model – such as, perceived distance as obstacle to accessing care, etc. Does the dataset contain variables that measure actual distance to the health facility? If not, I would consider removing these variables from the model.

- There is insufficient justification for including antenatal visits as “need” based characteristics. Within Anderson’s model, the construct of ‘need’ refers to one’s perception of their health needs. It might be useful for the authors to clarify how the number of antenatal care visits a woman has had appropriately captures this construct.

- Have the authors considered including controls for social identities, such as religion and ethnicity?

- I’m assuming this data includes individuals from all regions of the country? Why hasn’t region been added as a control in the model? There may be trends/patterns in utilization of maternal health services by region, and this might be interesting to investigate.

- It might be useful to include a variable on pregnancy complications, if available within the dataset. Having a pregnancy complication is often an important determinant of accessing maternal health services.

Results and discussion

- The discussion section could be strengthened by elaborating on some of the initiatives taken by the Myanmar government to improve access to maternal health services.

- Line 267, this is an interesting discussion on the political climate within the country and the civil disobedience movement. Perhaps the authors could have a separate paragraph elaborating on this, and how it influences utilization of maternal health services.

6. PLOS authors have the option to publish the peer review history of their article (what does this mean?). If published, this will include your full peer review and any attached files.

Reviewer #1: **Yes: **Gebremichael, Shewayiref Geremew

Reviewer #2: No

---

## [Author Response · Author response to Decision Letter 0]

11 Mar 2022

Dear editors and reviewers,

Please find attached our revision of the manuscript PONE-D-21-29455 entitled, "Determinants of institutional maternity services utilization in Myanmar". We would like to thank you and the reviewers for your suggested revisions. Based on these suggestions, we have made substantial changes to this manuscript. The revisions are in Track Changes, and we have provided a detailed explanation below of these changes in relation to each reviewer’s comments.

Reviewer # 1

o Page 16/28: line 295: “multivariate analysis” … bivariate analysis

o Abbreviations list: Please include abbreviation list

Thank you very much for your suggestion to add the abbreviation list for this study. We have added the abbreviations list after the reference page (see pages 31-32, lines 658-662).

Abbreviations

ANC: Antenatal Care; aOR: Adjusted Odds Ratio; CI: Confidence Interval; MCH: Maternal and Child Health; MDHS: Myanmar Demographic and Health Survey; MMR: Maternal Mortality Rate; RHC: Rural Health Center; SRHC: Sub-Rural Health Center; UNICEF: United Nations Children's Fund; WHO: World Health Organization

Reviewer # 2

Abstract

1. Please note the direction of the effect of household wealth, education and husband’s occupation on institutional delivery.

As suggested, we have added the direction of independent variables on institutional delivery (see page 2, lines 19-23). 

The added sentence is “Women live in rural areas, states/regions with a high levels of poverty, poor households, experience with financial burden and the husband’s occupation in agriculture or unskilled labor were negatively associated with institutional delivery. While a greater number of ANC visits and level of the couple’s education had a positive association with institutional delivery”. 

Background

2. The introduction has been well-written with a focus on global maternal mortality, followed by narrowing down to Myanmar. However, in line 71, there is a switch to discussing barriers to accessing maternal health services among the rural population. It might be useful to provide some context here, such as, information on maternal mortality rate among rural women, disparities in utilization of maternal health services between rural and urban areas. If one of the main aims of this study is to understand disparities in access to maternal health services between rural and urban areas, it might be helpful for the authors to use this paragraph to highlight that aspect. 

We have revised the paragraph in the background section to show the mortality differential between urban and rural populations for this study (see pages 5-6, lines 76-85).

The added paragraph is “The maternal mortality level of the rural population in Myanmar was higher than their urban counterparts (310 vs 193), according to the 2014 Myanmar population and housing census [8]. In 2017, 76% of all maternal deaths occurred in rural areas and 23% occurred in urban areas [16]. As high as 90% of women in urban areas received at least one ANC exam by a skilled health care provider, compared with a maximum of 78% for women in rural areas [11, 12, 13]. The utilization of institutional delivery in urban areas was 51% in 2007 [11], 65% in 2010 [12] and 70% in 2016 [13]. In rural areas, the utilization of institutional delivery was 15% in 2007, which is more than three times lower than that of urban areas [11], although the level had increased to 25% by 2010 [12] and to 28% by 2016 [13].”

Methods

3. Line 135, it is unclear why only married women were included in the sample. 

It is our mistake. We corrected “only married women” to “ever-married women” (see page 9, line 160-161). 

The correction is “…after considering these four criteria, 3,642 ever-married women were included…”

4. Line 172, there is insufficient justification for why age and women’s occupation were chosen to be removed from the analysis. For instance, why would the authors not remove parity and husband’s occupation? Age is an important pre-disposing factor and determinant of institutional delivery.

Thank you very much for your suggestions.

1. We added age of the woman at last delivery and woman’s occupation in the analysis. This is because the age and women’s occupation are important pre-disposing factors in this study.

2. We dropped the parity variable from the analysis. 

(See page 8, line 142-143; page 11, lines 213-214 and 216; pages 14-15, Table 1).

5. As a reviewer, I am slightly skeptical of the inclusion of perception-based variables within the model – such as, perceived distance as obstacle to accessing care, etc. Does the dataset contain variables that measure actual distance to the health facility? If not, I would consider removing these variables from the model.

We totally agree with your concern. The major reason that we kept these variables in our analysis is because our study is based on Anderson’s Model, in which access to health care service is a multidimensional function of health service accessibility. Aday and Andersen (1974) divided accessibility into two aspects: Socio-organizational and geographical access. Distance, transportation, travel time, and associated cost are included in the geographical access variable. 

The Myanmar DHS (MDHS) 2015-16 did not collect data on distance to the health facility or cost for treatment. However, there are questions which asked whether respondents faced problems related to the “distance to the health facility”, and “having enough money for advice or treatment”. In this study, we decided to use these two variables as proxies for “distance to health facility” and “affordable cost,” respectively. The MDHS 2015-16 report (page 127) also used these two variables in explaining health service accessibility in Myanmar [13]. 

6. There is insufficient justification for including antenatal visits as “need” based characteristics. Within Anderson’s model, the construct of ‘need’ refers to one’s perception of their health needs. It might be useful for the authors to clarify how the number of antenatal care visits a woman has had appropriately captures this construct.

We have added the specific clarification of the construct of the ANC visit in “need” based characteristics (see pages 7-8, lines 133-139). 

The added paragraph is: “Women who perceive the need for professional help and are aware of the risks of pregnancy and delivery are expected to seek antenatal care (ANC) and prepare for delivery in advance. Risk assessment could be conducted and women may be advised to deliver at the health facility if abnormal conditions of pregnancy are found during the ANC visit, such as twin pregnancy. Having four or more ANC exams may reflect a woman's concern about her pregnancy, experience of danger signs of pregnancy, and the need for professional help [28-30].”

7. Have the authors considered including controls for social identities, such as religion and ethnicity?

We are aware of the importance of religion and ethnicity in a woman’s consideration of utilization of institutional delivery. Other studies have found that ethnicity and religion are closely linked to cultural norms, and are thought to impact beliefs and values related to childbirth and health service utilization [29]. We have reviewed the MDHS dataset and there is no data for these variables (i.e., religion and ethnicity). Therefore, we added this condition in the ‘Limitations’ (see page 22, lines 434-436).

The added sentence is: “Ethnicity and religion are closely linked to cultural norms and are thought to impact beliefs and values related to childbirth and health service utilization [29]. Nevertheless, these variables are not available in the MDHS dataset.”

8. I’m assuming this data includes individuals from all regions of the country? Why hasn’t region been added as a control in the model? There may be trends/patterns in utilization of maternal health services by region, and this might be interesting to investigate. 

Thank you very much for this valuable suggestion. We added this control variable in our analysis. Myanmar is comprised of seven regions (Tanintharyi, Sagaing, Magway, Bago, Yangon, Mandalay, Ayeyarwady), seven states (Kachin, Kayah, Kayin, Chin, Mon, Rakhine and Shan), and one union territory (Nay Pyi Taw). (See page 5, lines 73-75). 

The added paragraph is: “There were 13,594 village tracts located in seven states (Kachin, Kayah, Kayin, Chin, Mon, Rakhine and Shan), seven regions (Tanintharyi, Sagaing, Magway, Bago, Yangon, Mandalay, Ayeyarwady), and one Union Territory (Nay Pyi Taw).” 

The result of the analysis is presented on page 11, lines 209-210; pages 12-13, lines 239-254; and pages 14-15, Table 1.

9. It might be useful to include a variable on pregnancy complications, if available within the dataset. Having a pregnancy complication is often an important determinant of accessing maternal health services. 

We agree and have added the experience of pregnancy complications as a variable in our analysis. The MDHS data did not have a direct question on pregnancy complications. Therefore, we used the data from a question that asked about previous experience of pregnancy termination as a proxy variable of pregnancy complications (see page 15, lines 19-20). 

The result of the analysis is presented on page 11, lines 213; pages 14-15, Table 1; and page 18, lines 341-347.

The addition to the discussion paragraph is: “Women who experienced pregnancy complications may have high utilization of institutional delivery. A study conducted in Nepal found that women who reported pregnancy complications such as bleeding had a high odds of institutional delivery [24]. Women may experience more complications of pregnancy as they get older. Similarly, multiparous women may also experience a higher prevalence of pregnancy complications. Consequences of previous births or miscarriages may make women more aware of the risks of childbirth and the benefits of delivery at a health facility [29]”

Results and discussion

10. The discussion section could be strengthened by elaborating on some of the initiatives taken by the Myanmar government to improve access to maternal health services.

Thank you very much for this suggestion. In the Discussion section, we added the health care activities implemented by the Myanmar government to improve maternal health care (see pages 20-21, lines 387-400).

The added paragraph is: “In the past, MOHS concentrated the MCH and Reproductive Health issues under the five-year Strategic Plan for Reproductive Health (2004-2008, 2009-2013, and 2014-2018) by providing easily accessible to modern contraception, increasing skill health service providers, and promoting ANC services for pregnant women. In 2017, maternal death surveillance and response systems were implemented in every state and region with the purpose of reducing maternal mortality by improving the quality of care and investigating the causes of maternal mortality [16]. The 2017-2021 Strategies towards Ending Preventable Maternal Mortality (EPMM) in Myanmar was approved in 2018. In 2020, MOHS produced “the clinical management guideline for COVID-19 infection in pregnancy”, in which all health care units that provided maternal health care services deliver a hotline or contact number to address patient concerns or inquiries [57]. At present, the 2021-2025 National Strategic Plan for sexual, reproductive, maternal, newborn, child and adolescent health (SRMNCAH) and action plans were scheduled to begin in the first quarter of 2021, delayed due to unstable political crisis [58]”

11. Line 267, this is an interesting discussion on the political climate within the country and the civil disobedience movement. Perhaps the authors could have a separate paragraph elaborating on this, and how it influences utilization of maternal health services.

We have added a discussion about the current political situation and CDM effect on utilization of healthcare services (see pages 21-22, lines 401-425).

The added paragraph is: “However, in 2021, most civil servants, including government health professionals, participated in the Civil Disobedience Movement in which healthcare professionals protested by not attending work since the February 2021 coup. The immediate impact was insufficient staffing, and impaired ability to provide essential health care. Therefore, the accessibility of healthcare services in Myanmar is deficient [59]. Although the Ministry of Health and Sports (MOHS) had urged healthcare professionals to resume their responsibilities, it is unclear how many of them have returned to work [58]. Moreover, the armed conflict has continued in Kayah, Chin, Shan, and the Kachin States, and Sagaing Region that has created thousands of internally displaced persons (IDPs), including pregnant women and newborns [60]. World Health Organization has supported MOHS in the implementation of early warning, alerts, and response system (EWARS) for IDPs via mobile clinics in conflict-affected areas, especially in Kachin and Rakhine States. The service implementation areas declined from 57 locations with 7,998 consultations to 24 locations with 1,282 consultations in the three months after the coup [58]. In Chin State, one pregnant woman, two newborns, and three elderly people died due to a lack of healthcare services while fleeing during the fighting [61]. 

After the coup, the curfew (10:00 PM to 4:00 AM) restricted the movement of people, and that posed a major obstacle for emergency obstetric referral service [58]. The curfew also certainly adversely affected accessibility to ANC, delivery, and reproductive health services due to the interruption of the transportation schedules for required supplies at health facilities. What is more, Myanmar's gross domestic product (GDP) fell by 18 percent in the 2021 Fiscal Year, and the economy has deteriorated due to the dual shocks of the coup and the Covid-19 pandemic. The main breadwinners of many households have lost their jobs, or are suffering from reduced income and high commodity prices [62]. These adverse effects are sure to worsen in the months ahead, and that will continue to negatively impact the utilization of health care services and, in particular, institutional delivery”

Additional References

Aday LA., Andersen R. A framework for the study of access to medical care. Health services research. 1974; 9(3): 208.

---

## [Editor Report · Decision Letter 1]

16 Mar 2022

Determinants of institutional maternity services utilization in Myanmar

PONE-D-21-29455R1

Dear Dr. Lwin,

We’re pleased to inform you that your manuscript has been judged scientifically suitable for publication and will be formally accepted for publication once it meets all outstanding technical requirements.

Kind regards,

Kannan Navaneetham, PhD

Academic Editor

PLOS ONE
---

## [Editor Report · Acceptance letter]

14 Apr 2022

PONE-D-21-29455R1 

Determinants of institutional maternity services utilization in Myanmar 

Dear Dr. Lwin:

I'm pleased to inform you that your manuscript has been deemed suitable for publication in PLOS ONE. Congratulations! Your manuscript is now with our production department. 

Kind regards, 

on behalf of

Prof. Kannan Navaneetham 

Academic Editor

PLOS ONE